# Research Paradigm of Network Approaches in Construction Safety and Occupational Health

**DOI:** 10.3390/ijerph191912241

**Published:** 2022-09-27

**Authors:** Mei Liu, Boning Li, Hongjun Cui, Pin-Chao Liao, Yuecheng Huang

**Affiliations:** 1School of Urban Economics and Management, Beijing University of Civil Engineering and Architecture, Beijing 100044, China; 2Department of Construction Management, Tsinghua University, Beijing 100084, China

**Keywords:** network approaches, construction safety management, occupational health, bibliometric analysis

## Abstract

Construction safety accidents seriously threaten the lives and health of employees; however, the complexity of construction safety problems continues to increase. Network approaches have been widely applied to address accident mechanics. This study aims to review related studies on construction safety and occupational health (CSOH) and summarize the research paradigm of recent decades. We solicited 119 peer-reviewed journal articles and performed a bibliometric analysis as the foundation of the future directions, application bottlenecks, and research paradigm. (1) Based on the keyword cluster, future directions are divided into four layers: key directions, core themes, key problems, and important methods. (2) The network approaches are not independently applied in the CSOH research. It needs to rely on different theories or be combined with other methods and models. However, in terms of approach applications, there are still some common limitations that restrict its application and development. (3) The research paradigm of network analysis process can be divided into four stages: description, explanation, prediction, and control. When the same network method encounters different research objects, it focuses on different analysis processes and plays different roles.

## 1. Introduction

Construction safety gravely reflects the well-being of society because workplace accidents incur various losses to injured workers and their families, employers, and society [1]. Considerable efforts have been made to protect workers’ occupational health and improve construction safety performance. However, construction projects involve complex collaborations between multiple stakeholders and significant workplace changes [2]. Most construction activities are carried out in a rapidly changing environment and under changing site conditions, and are characterized by complex construction techniques, cross-team operations, heavy manual labor, and high mobility of construction workers. Many unsafe factors of construction with internal interaction have been identified, including working conditions, working conditions in hazardous or unsafe environments, hazards, and health and safety problems of workers [3], making construction one of the most dangerous industries and a complex management project [2,4].

In recent years, there have been an increasing number of publications on construction safety and occupational health (CSOH) from different countries, such as the United States, China, Australia, and England. Traditional local subjective qualitative reasoning and static linear observations are not sufficient to deal with the complexity of current construction projects. Modeling construction projects and their subcategories into a network that allows multiple types of nodes and accurately defining and identifying the relationships between them can help deepen the understanding of construction project safety and improve management strategies. The continuous development of network science provides researchers with numerous network analysis methods and tools and allows the combination of various theories to deepen research on construction project management. Network theory has been widely adopted to reveal and describe complexity, including uncertainty, randomness, ambiguity, and other properties [5]. Complex networks exist in human social systems [6]. Systems can be efficiently modeled using a network structure in which the system entities are network nodes and the relations between the entities are network links [7]. For example, the dynamic network approach enables researchers to dynamically observe the interdependence between risks [8]. It helps overcome the limitations of traditional research, which relies only on qualitative reasoning to identify and define the coupling relationship between safety hazards, without consideration of the dynamic characteristics of construction projects, thus resulting in the lack of universality in current research. Moreover, it contributes to the implementation of strategic and forward-looking safety management during construction.

Network analysis methods and technologies are often used to describe complexity and observed phenomena by modeling and constructing a framework to represent the relationship and mutual influence. Based on the description, quantitative calculation and qualitative analysis are employed to analyze and explain, so that the prediction can be realized on the premise of understanding the mechanism to improve safety management performance. In addition, some network-based technologies, such as sensor networks, are used to assist safety management and provide valuable information. Based on these criteria, the literature can be divided into four categories: description, explanation, prediction, and control. This is not only conducive to comprehensive understanding of the frontier of network analysis methods and technologies in CSOH research, but is also conducive to the future development of network research in this field. It should be noted that since there may exist studies with multiple research purposes, for example, some studies involve both description and explanation or both explanation and prediction [9]. Thus, in the following literature classification process, articles were allowed to correspond to multiple purposes. However, most previous review studies have only focused on the application of a certain type of network approach, and the scope of the research is not precise, which makes them insufficient to provide insights for the development of construction safety research. For example, the critical review conducted by Lee et al. [10] clarified the application of social network analysis to complex project management.

Therefore, the research questions are as follows:Q1: What objectives, methods, and theories are focused on in the application of existing network approaches in CSOH and its future direction?Q2: When applied in CSOH, what are the bottlenecks in the construction process of the network model in terms of data character and application and verification in engineering practice?Q3: What is the network analysis paradigm from the perspective of description, explanation, prediction, and control?

This study responds to the need for a better understanding of the potential applications of network approaches in construction safety and occupational health management (NA-CSOHM). Within this context, this paper aims to provide a detailed review of research on network approaches to delineate their application areas, identify research gaps, and guide potential research directions in CSOH. The remainder of this paper is devoted to the methodology, an overview of the literature based on statistical and bibliometric tools, discussions, challenges, and future directions for research and conclusions.

## 2. Methodology

### 2.1. Literature Retrieval

Web of Science was used to search and select articles related to the targeted scopes because the abundant quantity of qualified academic papers it includes is adequate to represent the main body of the network approaches in construction safety and occupational health management (NA-CSOHM). To ensure that all relevant literature was captured through the literature retrieval process and to conduct a comprehensive review, different unique search keywords describing the subject matter were required. The Boolean operators “AND” and “OR” were used as logical operators to combine and formalize keywords. Three sets of keyword terms were used to triangulate articles related to NA-CSOHM, network approaches, and construction projects, as shown in Table 1.

The specific strategy of literature retrieval in this review was to take the intersection of the search results of Groups 1, 2, and 3. An advantage of this search strategy is that sufficient and comprehensive relevant literature can be obtained. The above operations are carried out on the Web of Science and take advantage of the “AND” and “OR” options in its “combine set” function to complete the process of taking unions and intersections. To provide a broad review of the target areas, considering that conference papers may propose the latest interesting methods and indicate the application prospects and possibility of new methods, we did not exclude published conference papers.

According to the search keywords listed in Table 1, 1586 articles were obtained from the WOS core collection. Considering the probability that papers with little relevance were listed in the search results, it was necessary to eliminate unwanted ones. To limit the scope of the search results, we conduct a filtering process by evaluating the search results. In the first step, only journals and conference papers from 2000 to 2022 were saved, leaving 1524 articles. In the second step, the WOS categories and publication titles were carefully examined using the WOS screening strategy, and papers that were irrelevant to the construction industry were excluded, resulting in 187 articles. By screening the theme and research field, the fit and quality of the obtained research literature are guaranteed. Moreover, the paper resources collected from the Web of Science Core Collection database were sufficient and highly reliable, as papers were selected after rounds of strict peer evaluation and selection. Because the literature analysis implemented in this study relied on the title, abstract, and keywords of the paper, the methodology, emphasis of the research field, and findings were all included. The results were highly accurate in terms of the precision of the extracted information.

Finally, the less relevant and irrelevant papers were screened after a brief visual examination of their contents, leaving 119 publications for further analysis. The two criteria of the content examination in the second literature filtering process were as follows: (1) focus on the field of CSOH; and (2) adopt network approaches as the main research methods or introduce network approaches with the prospect of application in CSOH management.

### 2.2. Bibliometric Analysis

A manual review of large-scale papers is impractical because understanding the development and trends of several fields and their intersection requires analyzing a large number of papers, extracting information and knowledge, and outlining the framework of the current NA-CSOHM field of study. Moreover, bias, subjective categorization, and judgments are prone to human-related systematic errors and false analyses [11]. The bibliometric approach is used as the primary analysis method, which integrates the information of large-scale data by scientific calculation and allows for an examination of existing literature based solely on reported data [12]. This method can help minimize potential author bias and map the structure and evolution of knowledge. Through visualization of the findings, the relationship between study fields can be accurately grasped, and the landscape of abundant or rare resources can be clearly depicted. The software CiteSpace 6.1.R3 (Philadelphia, PA, USA) developed by Chen [13] was adopted as the tool for the analysis in this study, and it generated bibliometric networks as well as mapping visualization based on the selected 125 articles.

### 2.3. Keyword Cluster Analysis

Scrutinizing the information keywords may be crucial for understanding development trends in the NA-CSOHM field. However, relying only on scattered keywords cannot clarify the main fields and research structure, because the keywords are too detailed and contain limited information. Cluster analysis was conducted through mathematical and statistical calculations of the text data of keywords and abstracts to summarize potential semantic topics and their overlap [10]. Through multidimensional clustering analysis, the study of merged concepts with strong internal connections can help understand the prevalent thoughts and frontiers in this field. Cluster analysis employs a set of algorithms to attract and repel word links and then classifies the cluster groups [14]. Each cluster reserves and represents a certain number of observations. Researchers can determine which algorithm will be used to name the cluster in CiteSpace. In this study, the log-likelihood ratio (LLR) algorithm, the centrality of keywords, was adopted to name the clusters and understand the meaning of clustering and the connections between groups [10].

## 3. Results

### 3.1. Distributions and Trends

#### 3.1.1. Distribution by Year of Publication

The results obtained from the bibliometric search demonstrated trends in the research on this topic, as shown in Figure 1. The emergence of the first publication in 2006 marked the point at which researchers began to apply network approaches and technology to construction safety management. Since 2012, research in this field has attracted increasing attention from scholars and has gradually become a popular research topic. In total, 24 articles were published by 2022, making this the most significant year for this emerging field. Overall, there is an increasing trend in the adoption of network approaches in the CSOH research. In the early stages of research in the field of NA-CSOHM, scholars mainly focused on attempts to apply network approaches to construction project management, such as risk and cost management in construction projects, to improve performance in these aspects. Network approaches have been applied to more specific research in later developments such as safety culture, safety climate, injury and accident prevention, behavior and human error, and factor classification studies. Social networks and Bayesian networks are the two most used network approaches, and are widely used to describe, explain, and predict various scenarios and systems of construction projects. With the development of information technology and computer science, theories and technologies of various other disciplines have more opportunities to be combined with network approaches to improve the management capacity of construction projects. For example, a combination of BIM and network technology can help achieve accurate management across the phases of a construction project [15]. The development of Internet of Things (IoT) technology allows network technology to be used to assist in the comprehensive management and control of various types of equipment and machinery on construction sites [16].

#### 3.1.2. Distribution by Countries and Institutions

To reflect the performance of research institutions and countries in this field, we conducted statistics on the authors of the literature. The differences in publication quantities of various countries imply the extent to which research commitment and value are observed in these countries.

According to the statistical results in Figure 2, 27 countries have conducted research in this field. China, the United States, Australia, England, and Singapore are among the top five countries that have led research in this field. It indicates that the large size of the construction industry and the large number of construction workers prompted researchers in the country to focus on the health and safety of workers engaged in the construction industry and to actively explore feasible safety management methods and strategies.

Table 2 shows the number of institutions that have published three or more papers, among which the Huazhong University of Science and Technology has published the most relevant research, mainly focusing on subway construction, intelligent perception and pre-control. Hong Kong Polytech University, Tsinghua University, and the University of Maryland have also paid close attention to human error and fall prevention in CSOH.

#### 3.1.3. Distribution by Journals

Figure 3 shows the distribution of articles published in major journals. The top six major NA-CSOHM journals accounted for 71 articles (59.66%), 29 of which were from *Safety Science* (24.37%), 11 from the *Journal of Construction Engineering and Management* (9.24%), 10 from *Reliability Engineering & System Safety* (8.40%), 8 from the *International Journal of Environmental Research and Public Health* (6.72%), 7 from *Engineering Construction and Architectural Management* (5.88%), and 6 from *Automation in Construction* (5.04%). This distribution shows that research in the NA-CSOHM field has gained attention from high-quality journals in the fields of construction project management and safety science. Constant updating and development of network approaches and technologies may also involve many emerging technologies, such as sensors. As one of the significant application scenarios of these emerging technologies and methods, CSOH has received increasing attention owing to its great social welfare impact.

### 3.2. Keyword Co-Occurrence Analysis

To identify the research hotspots in the field, we used CiteSpace to analyze the keywords in the literature. The keyword co-occurrence information is shown in Table 3. Keywords are sorted by occurrence frequency, and only keywords with a frequency greater than 3 are listed: co-occurrence keywords list 5 and the corresponding journal list 3 according to co-occurrence frequency.

The core domain of this research is “safety management,” “construction safety” and “accident analysis.” The hotspots of methodology include “Bayesian network,” “complex network” and “artificial neural network,” etc. This result indicates that (1) the network approach is mainly used to improve the performance of construction safety management; (2) the Bayesian method is the most used network approach; and (3) network methods are often used to describe complex and interactive systems and events by modeling or building frameworks, such as describing human error, behavior, accident causation, and construction of a risk association network.

### 3.3. Keyword Cluster Analysis

Figure 4 shows clusters that are numbered in descending order of cluster size, and the naming of clusters is conducted automatically using CiteSpace. The clustering results are #0–#7, including 8 categories. As shown in Figure 4, each cluster consists of multiple closely related keywords. The smaller the cluster number, the larger the cluster size (more keywords). Modularity Q value greater than 0.3 indicates significant clustering structure; Silhouette S value greater than 0.5 indicates that the clustering is reasonable, and greater than 0.7 indicates that the clustering is convincing. Q = 0.7972 and S = 0.9128 indicate that the keyword clustering map is reasonable and reliable, and the map can show the main fields of current related research. The size, S value, and top terms of the eight cluster results are shown in Table 4.

Eight clusters were identified and named based on the keywords shown in Figure 4 and Table 4. Cluster 0 indicates that the Bayesian network is the most popular and widely applied network approach in the early stages of study in the NA-CSOHM field. The establishment of a network generally requires the support of a large amount of available data, which makes data mining technology of great significance to the development and application of network approaches and has gradually gained more attention [17]. Clusters 3 and 7 illustrate that the application of various information technologies and new hardware technologies, such as wireless sensors and the Internet of Things (IoT), in CSOH management research has gradually formed a trend and has broad prospects. The future development of new technologies in various disciplines will enable network approaches to continuously derive more application scenarios and possibilities for safety management. Cluster 1, “construction safety,” and cluster 4, “safety management,” indicate that the main purpose of this research that adopted network approaches in CSOH management is to improve the safety management of the construction industry. This clustering result also affirms the quality of the literature retrieval, because the subjects of these studies were consistent with the research scope of this review. Cluster 2, “human factors,” preferred to use holistic approach for accident analysis. From cluster 5, “proactivity,” and cluster 6, “construction sites,” immediate actions and comparative analysis are persistent concerns in CSOH future orientation.

**Table 4 ijerph-19-12241-t004:** Keyword cluster information.

Cluster ID	Size	Silouette	Top Terms	Author(s), Year of Publication
#0 bayesian network	28	0.954	bayesian network; bridge construction; fall risks; fault tree; risk assessment	[18,19,20,21,22,23,24]
#1 construction safety	21	0.904	construction safety; smartphone; sensitivity analysis; dynamic bayesian network (dbn); convolutional neural networks	[25,26,27,28,29,30]
#2 human factors	19	0.991	human factors; safety assessment; holistic approach; safety engineering; accident analysis	[31,32,33,34,35]
#3 data mining	18	0.935	data mining; fuzzy anp; fuzzy fmea; labor and personnel issues; neural network	[36,37,38,39,40]
#4 safety management	16	0.898	safety management; complex network; subway construction; hangzhou subway construction collapse (hzscc); visibility graph	[41,42,43,44]
#5 proactivity	15	0.801	proactivity; future orientation; comparative analysis; unsafe behaviors; occupational safety	[18,45,46,47,48]
#6 construction sites	14	0.86	construction sites; construction management; safety; immediate actions; risk management	[21,38,49,50,51]
#7 artificial intelligence	13	0.899	artificial intelligence; accident management; generative adversarial network; automation; dynamic probabilistic risk assessment	[23,52,53]

### 3.4. Network Approaches, Research Objects, and Analysis Process

The main network approaches applied in CSOH are listed in Table 5. As this study focuses on the application paradigm of network approaches, the calculation process for each network approach is not explained here. According to 119 documents, the application process of network research involves four aspects: description, explanation, prediction, and control, as shown in Table 5. For example, Zhou et al. described subway construction accidents by building a network model and presented a network interpretation of the rapid propagation speed between subway accidents by calculating and analyzing network elasticity, small world, and other characteristics reflected by topology parameters [41]. The distribution characteristics of the shortest path of the network can be used not only as the basis for accident prediction but also as the main strategy for controlling the chain reaction and propagation rate in the accident network. However, owing to the differences in research objects and research objectives, the importance of the above four aspects in different studies is slightly different, which needs further discussion.

This indicates that different network approaches can be applied to various research objects. As the most frequently used network methods, Bayesian networks (BN and DBN) can be used in almost all engineering projects, such as electrical and mechanical, bridge construction, steel construction, hydraulic engineering, and tunnel construction. At the same time, they are also used to explore the relationship between unsafe behaviors and their influencing factors in human errors, labor, and personal issues. BN analysis focuses more on the description, explanation, and control measures of accident risk-influencing factors. A DBN can conduct a predictive analysis by simulating the risk transmission mechanism. Neural networks (NN, ANN, FNN, and CNN) were applied to analyze the safety atmosphere, occupational health and safety, and safety behavior in construction safety management, especially in fall prevention. Because NN is a way to realize machine learning, it is more suitable than other network methods for explaining the current situation and predicting the future. SNA has also been gradually popularized and applied in the fields of sociology and management to describe the risk correlation and transmission mechanisms of engineering construction. Based on the analysis of the overall and individual network structures, SNA can describe and explain the relationship between risks and unsafe behaviors and their influencing factors. Simultaneously, dynamic prediction can be realized by combining other models, such as the cascading failure model and BIM.

## 4. Discussion

### 4.1. Future Direction Based on Keyword Cluster Analysis

Table 4 shows the clustering results for 119 papers, and eight clusters were obtained. From the perspective of theme words of clustering, the above eight clusters have hierarchical characteristics from macro to micro, which can be divided into four layers: key directions, core themes, key problems, and important methods.

Cluster #1, “construction safety,” is the key research direction with considerable scale, and it is also one of the core directions in the field of construction safety and occupational health (CSOH). From the size of cluster #1, “construction safety,” it is the second of the eight clusters, indicating that in the CSOH field, the network approaches are currently more used to study construction safety issues, while construction occupational health is given less attention. From the key keywords in cluster #1, the Bayesian network is the most widely used and is mainly used to build a more precise and accurate accident and safety risk assessment method [27,28]. However, in research on the identification of hazards and risks, an increasing number of scholars have realized that traditional risk identification methods relying on manual methods have serious deficiencies in timeliness. With the popularization of information technology in the construction industry, the accessibility and scale of security risk-related data have been revolutionized, and methods such as artificial neural networks and convolutional neural networks have increasingly aroused the interest of scholars to build more accurate and efficient risk identification algorithms. For example, CNN has been used to build an image detection model for safety barriers [29], and the ANN method has been combined with the smartphone to identify near-miss falls [26]. Therefore, diversified network methods have shown considerable application potential in the research of CSOH fields and will also have a wide range of application scenarios in the future when dealing with the problems of organic health in construction projects. Moreover, the information and intelligent development trend of the construction industry, which aims to improve efficiency and quality, has also increased the demand for network methods.

Cluster #4, “safety management,” has gradually become the core theme of construction safety research based on network methods, showing the applicability of network methods in management research. In recent years, the previous understanding of the simple chain and linear nature of accidents and risks has had limitations, so the complexity of safety management has gradually attracted the attention of scholars [43]. Based on the keywords of cluster #4, a complex network is introduced in the research on safety management, which helps to describe the complex phenomena of accidents and risks. Particularly in characterizing the research objects composed of a large number of elements, multiple levels, complex interactions, etc., the node and edge definition of the network method can more clearly represent the accident system, risk system, management system, and other complex safety research objects. For example, a complex network can be used to extract various risk factors and their complex triggering relationships from construction accidents, and several weighted risk networks can be generated according to different accident types with the accident level and frequency as weights [44]. Zhou et al. studied the complexity of a subway construction accident network (SCAN) based on complex networks [41]. This indicates that the network method has become an important method in construction safety management research. However, several complex problems remain in the field of CSOH. In future research, the transformation of complex research objects into corresponding network models to achieve a more reasonable representation is an issue that needs further exploration. In addition, because the network method itself is accompanied by a wealth of analysis parameters and tools, the actual safety connotation reflected by the parameter levels obtained from the perspective of topology structure is another issue that needs to be addressed.

Clusters #2, “human factors,” #5, “proactivity,” and #6, “construction sites,” reflect the core themes in the research on network methods in CSOH. Human safety management is an important and challenging problem in terms of construction safety. Unsafe human behavior has been identified as the highest and most important cause of accidents in many accident investigations and studies, and it is increasingly urgent to solve the human factors in safety problems in the process of continuous innovation of production technology to improve the unsafe state of objects. However, in the field of CSOH, unsafe human behavior is aggravated by the complexity and variety of influencing factors and interactions involved, and workers are still in the complex dynamic space of the construction site. The network method has been widely adopted in human error research, with the most important advantage in revealing the influence and evolution mechanics of human errors, thus providing a more accurate basis for improving unsafe behaviors and mistakes [32]. Therefore, with the expansion of human error research, network methods have systematic application needs to describe the relationship between factors and effects, reveal the mechanism of influence, predict the spread of influence, and formulate intervention measures. In addition, passive safety behaviors under the restraint of safety rules do not have consistent sustainability. Once the rules or executive power change, unsafe behaviors can rebound. For example, safety communication is one of the factors that promote the safety initiative of workers at construction sites. The social network method was applied to build a model of safety communication among workers and reveal a way to promote safety communication based on network topology characteristics [47]. Therefore, improving proactivity in safety and health management is an issue that requires more attention at present and in the future [45].

Clusters #0, “Bayesian network,” #3, “data mining,” and #7, “artificial intelligence,” are important research methods closely related to the application of network methods in CSOH. Compared with traditional safety analysis and evaluation methods such as ETA, FTA, and FMECA, the Bayesian network is mainly used to replace or improve traditional safety analysis and evaluation methods by virtue of its fine description of safety factors and their interactions, as well as its better quantitative performance [19,20,21]. Moreover, as the application scope of the Bayesian network increases, information technologies such as data mining and artistic intelligence are gradually being introduced into the research of the Bayesian network, which greatly improves the dynamics, accuracy, and timeliness of security assessment methods [23,24]. In the construction industry, artificial intelligence, big data, cloud computing, and deep learning have gradually been combined with research and practice, which not only shows a stronger analytical ability for the original security problems, but also plays an important role in finding and solving new security problems.

### 4.2. Network Application Challenges

According to Figure 1 and Table 5, the number of studies using the network approach in CSOH management research continues to increase, and the advantages of the network approach are evident. The development of network science provides a theoretical basis for the application of various network approaches in CSOH research, enabling the integration of network approaches and other theories. However, in terms of approach applications, there are still some common limitations that restrict its application and development.

#### 4.2.1. Theories Integrated in the Application of Network Approaches

According to Table 3, the “network theory” is the co-occurrence keyword with the high-frequency keyword “safety management,” and by intensive reading of the selected literature, the network approaches are not independently applied in the CSOH research. It needs to rely on different theories (complex network theory, accident causation theory, communication theory, etc.) or be combined with other methods(NLP, artificial intelligence, IoT, etc.) and models (attribute-based building safety framework, epidemiological models SEM, BIM, etc.) to describe, model, and even reconstruct accident patterns and the contents of safety management systems. These theories and models can help study the relationship and interaction behavior in the network of construction safety and occupational health management. Among them, the complex network theory and accident cause theory are the most used theoretical methods when using network methods for analyzing human error related to CSOH. Considering the attributes and data characteristics of the research objects, the network model built based on an attribute-based building safety framework provides a more flexible definition and objective measurement of the association between research risks.

The basic objective of safety research is accident prevention, which begins with a clear understanding of the factors that play a key role in causation [87]. In the field of CSOH, the complex network theory is suitable for describing risk analysis and accident causation. The combination of network approaches and accident causation theory has significantly improved the ability to accurately capture and describe the complexity of accidents. The accident causation theory provides a theoretical basis for defining nodes and connections in a network that are related to risks and hazards. The complexity of construction safety management is reflected in the process of different risk consequences caused by the direct or indirect effects of the risk factors. The effect of factors and factor associations on the system is not a simple addition of factors but mutual penetration and influence, and finally presents a nonlinear coupling relationship [88]. Based on network theory and accident causation theory, researchers have begun to focus on the dynamic propagation of risks in a network. For example, in previous studies, some scholars introduced the cascading failure analysis approach after establishing a hazard interdependence network, thus attempting to describe the propagation mechanism of hazards [32,89]. Complex network theory has unique advantages in describing risk propagation in CSOH because it helps to accurately express and quantify the mechanism of interaction. Therefore, based on the combination of complex network theory and accident causation theory, it is a potential solution for accident analysis and human error analysis by establishing an accident-risk network and introducing dynamic analysis approaches based on the topological characteristics of the network.

In addition, by defining “construction safety clashes” as incompatibilities among fundamental attributes of the work environment that contribute to construction injuries, construction accidents can be regarded as perturbations in gene regulatory networks that are composed of the basic attributes of the construction project [90]. The attribute-based building safety framework allows the extraction of standardized safety information from objective raw text data to construct a risk network. This method assumes that any event can be regarded as a result of the co-occurrence of basic attributes and the existence of workers. This unified approach allows the extraction of standardized safety information from objective and raw textual data such as injury reports. Tixier et al. developed a natural language processing (NLP) system with high accuracy to automatically scan and extract relevant safety attributes and their safety-critical associations [91]. Fundamental attributes are universal context-free descriptors of the jobsite, allowing it to span construction means and methods, environmental conditions, and human factors [92,93]. It is worth noting that the development of tools and technologies highlights the advantages of attribute-based building safety framework, which is suitable for building network with the analysis and utilization of unstructured data.

#### 4.2.2. The Bottleneck in Network Approach Application

When applied to CSOH, bottlenecks exist in the construction process of the network model in terms of data quality and calculation speed, and there is also a bottleneck in the application and verification in engineering practice.

From the view of quantity and quality of data, because the network contains more relationships (edges) than individuals (nodes), a large amount of data is required to construct the network [94]. If multiple networks need to be constructed, such as a multilevel network or a dynamic network, then each network should contain sufficient data to build a robust network and have similar (or equal) sampling work to avoid the possibility of an impact on the final network caused by the sampling methods. In addition, the data quality is a significant constraint. On the one hand, although a large number of data records are generated during the construction project management process, the proportion of available structured data that can be directly used to construct the network is small. Indeed, most information consists of unstructured data of various formations that are difficult to apply directly to construct networks. In addition, missing, incomplete, and incorrect data are very common, which also restrict the construction and accuracy of the network. Furthermore, network construction is a laborious and slow process that requires considerable investment, cost, and time. The need to update the network is significant, considering the dynamics and complexity of construction projects. According to the research of Liao et al., when using data learning to analyze hazards and construct a network, the complexity of the network is proportional to the square of the amount of input data, which means that the network construction process may be laborious and slow, and may be accompanied by considerable noise [38]. Therefore, attaching sufficient importance to the opportunities that the development of computer science and technology may bring to the application of network approaches can broaden the data sources and help overcome the defects of data quality and computing speed.

From the perspective of practice and verification, the conclusions drawn by most existing studies using network approaches lack opportunities for proof in practice, which is related to the above two factors. Research based on specific cases lacks universality. However, the heterogeneity between construction projects makes it almost impossible for the same network to obtain batch verification opportunities for a large number of projects in a short period. Additionally, construction projects that provide sufficient data are rare. The superposition of these factors makes it difficult to apply network approaches in large-scale CSOH management research to fully demonstrate its advantages and the significance of safety management practices. Therefore, identifying and developing subnetwork models that can be migrated and applied flexibly in different construction projects to expand verification and application opportunities to evaluate the effectiveness and universality of the proposed method fosters the practical application of the research results.

### 4.3. Network Analysis Paradigm from View of Description, Explanation, Prediction, and Control

As shown in Table 5, different network methods have unique characteristics for different research objects. However, when the same network method encounters different research objects, it focuses on different analysis processes and plays different roles. The network analysis process can be divided into four stages (description, explanation, prediction, and control), and the research paradigm under different objectives can be discussed.

#### 4.3.1. Description

Construction projects span a wide range of spaces, times, and organizations, involving complex interactions between people, machines, materials, and the environment. The network approach can clearly describe the relationships among a large number of nodes without the requirement for nodes to be objective entities. These three topics can be summarized according to the differences in the target objects.

The first research topic was the characterization and description of the static pattern and structure of the target network. Its main purpose is to identify and define the nodes in the network and the relationships among the nodes to improve the management capability and performance. Nodes can be entities or groups of stakeholders in a construction project or they can be indicators and factors that are important for project management, such as the relationship between risk-related networks, stakeholder networks, and factors that may affect safety.

The second topic is aimed at capturing, quantifying, and modeling dynamic propagation behavior in networks. Examples include dynamic risk propagation in a risk network and the interaction between risk clusters. Social networks are often used to describe the processes of information exchange, communication, and safety knowledge transfer in construction projects. Dynamic observations of the interdependence of risks are helpful for strategic and forward-looking safety management during the construction process.

The purpose of the third research topic is to describe and simulate the occurrence of incidents, such as risk-formation processes and safety-accident trigger patterns. For example, the attribute-based construction safety framework allows construction accidents to be regarded as perturbations and conflicts of the fundamental attribute network during the construction process, which provides a new perspective for researchers to view construction safety accidents. In addition, these studies are helpful for the early diagnosis of the safety performance of the construction process as they show the possible significant impact of some factors from a comprehensive and systematic perspective.

#### 4.3.2. Explanation

By describing and explaining the possible interrelationships between nodes in the network of a system, the network structure can be further optimized and adjusted, and critical nodes and important interactions can be identified. Explanation-oriented NA-CSOHM research can be divided into three stages.

The first stage involves screening and identifying the factors and risks that may affect the critical nodes and associated relationships in specific safety-related systems or processes—for example, the study of the factors that have a causal relationship with accidents, and how they influence or contribute to the outcome of accidents.

The second stage is to determine the priority and relative importance of the factors or conditions that impact the target event or phenomenon to define their weight or priority order. Many studies have attempted to quantify the systemic impact of risks, identify critical stakeholders, and prioritize management measures based on the interrelationships between risks. Theoretically, this helps researchers have a more systematic understanding of the causation of specific phenomena or incidents. Practically, this is beneficial for managers to formulate and improve safety management strategies more strategically.

The third stage is an explanatory study of the internal mechanism, influence mode, trigger path, and effect of the phenomenon, event, or system. Research at this stage may develop theoretical knowledge that can provide guidance for safety practices. For example, Li et al. studied how social capital affects the safety behavior of workers [95]. Garcia-Herrero used Bayesian networks to analyze the impact of various labor conditions on occupational accidents [96]. Feng et al. used neural networks to analyze the interactions between stakeholders in safety supervision [97]. Ma et al. developed a dynamic causal model of human errors that highlights the hazards that potentially trigger human error occurrences, thereby facilitating the implementation of proactive safety strategies and safety measures in advance [8].

The network approach can not only consider the static correlation between nodes but also consider factors such as time series and dynamic propagation. Such research is also enabled by evolving network technologies, such as meta-networks that allow for multiple layers of networks and complex networks that consider time series. However, owing to the time and space span of these networks, the construction process and exploration of the details of these networks require a relatively high data level, source, and quality. To a certain extent, this has also become a limitation in that it cannot completely replace expert opinions.

#### 4.3.3. Prediction

The best way to achieve good safety performance is to mitigate or minimize risk before it occurs [98]. Researchers have adopted various network approaches and tools to actively, comprehensively, and systematically predict and manage the safety performance of construction projects [99]. Predicted objects typically fall into two categories: safety incidents and safety indicators.

Accident prediction is usually based on the analysis of past accident cases and management data to identify the most influential factors or accident precursors among the factors that may have a causal relationship with the accident, establish an analysis model to calculate the accident probability, or predict the types of accidents that may occur [100]. For example, Ayhan and Tokdemir developed a new model that used latent category clustering analysis (LCCA) and artificial neural networks (ANNs) to predict the results of construction accidents and determine the necessary preventive measures [55]. Rivas et al. used data mining techniques (decision rules, Bayesian networks, support vector machines, and classification trees) to model accidents and incident data compiled from the mining and construction sectors to identify the most important causes of accidents and develop predictive models [9]. Gerassis et al. used a 6-year accident database based on data mining methods and attribute selection, and used Bayesian networks to quantify the specific causes of different types of accidents, thereby determining the main predictive factors and analyzing the key attributes of accidents associated with the construction of embankments [101].

Prediction research on safety indicators is generally specific, such as safety climate [54] and worker safety behavior [40]. By considering previous safety climate models, Zhou et al. proposed a Bayesian network (BN)-based model, establishing a probabilistic relational network among causal factors, including safety climate and personal experience factors, that influence human behavior pertinent to construction safety [58]. Patel and Jha developed a model to predict the safety climate in a construction project based on artificial neural networks (ANNs) to evaluate and differentiate construction projects based on their safety climate and determine the significant constructs of a safety climate [54]. Moreover, they further developed a model employing artificial neural networks (ANNs) to predict the safe work behavior of employees using 10 safety climate constructs determined through a literature review. The model utilizes safety climate constructs (determinants) as inputs and safe work behavior as an output to predict the safety behavior of employees, which helps effectively manage the safety of construction projects. Nguyen et al. proposed a Bayesian network-based high-altitude accident risk diagnosis method, predicting the safety risk of falling from a height and contributing to the construction of a building safety knowledge system [50]. Jahangiri et al. proposed a method that combined an adaptive neural network fuzzy inference system with a safety checklist to identify risk factors and predict the risk of falling from a scaffold at a construction site [102].

#### 4.3.4. Control

Traditional construction management requires a large human resource investment, characterized by high cost and uncertainty, and it is difficult to ensure efficiency and accuracy because of the complexity of the organization, space, and time. The application of the network approach allows researchers and managers not only to observe and search for possible safety vulnerabilities from a systematic and comprehensive perspective, but also to organize a safety risk monitoring system by integrating multiple entity monitoring devices and technologies to support the achievement of the “control” purpose and improve safety management efficiency [103,104].

The monitoring network on a construction site is mainly concerned with the unsafe behavior of people and the unsafe physical state of things, which are the main sources of potential hazards leading to safety accidents. The monitoring of human behavior mainly focuses on unsafe movements and location changes [105]. For example, Fang et al. developed an automatic computer-vision approach that utilizes a mask region-based convolutional neural network (R-CNN) to detect individuals traversing structural supports during the construction of a project [106]. Yang et al. integrated ZigBee technology and radio frequency identification (RFID) technology into a wireless sensor network, solved the problem of real-time identity information tracking, and contributed to the proactive prevention of accidents and the improvement of construction site safety [107]. Huang et al. proposed a smart risk perception system for risk sensing incorporating wireless sensor network (WSN) on-site visualization techniques and a resilience-based repair strategy, which can achieve real-time monitoring of geo-structural performance and dynamic pre-warning for the safety of on-site workers [108].

Network approaches and technologies not only allow multiple monitoring devices to be integrated as nodes to transmit and process monitoring information but can also simulate the decision-making process of experts based on a hybrid data fusion model of multi-source information to automatically conduct safety risk assessment and early warning. This also contributes to the establishment of an information collaboration network platform to replace traditional manual management, which may significantly improve the performance of information collection, sharing, and communication in construction projects.

### 4.4. Limitations of the Study

Network approaches have been widely applied to address accident mechanics, and this study aimed to review related studies on construction safety and occupational health (CSOH). However, in terms of the study of construction safety or occupational health alone, the application of network methods is more extensive and diverse, such as machinery failure monitoring, project quality monitoring, and the occupational health of manufacturing workers. However, the above is not included in this retrieval, because this study aims to focus on personnel safety and occupational health in construction projects in order to judge whether the introduction of network analysis methods can improve safety performance and workers’ safety to a certain extent, and then condense the research paradigm of network analysis methods. Therefore, when determining the search keywords in the literature retrieval and further manual screening, this study considered both “construction safety” and “occupational health.” However, it is still possible that the keywords and abstracts of one article only include “construction safety” and do not involve “occupational health,” but in fact, the result of this study contributes to occupational health in the text, and this kind of article was not extracted in the literature retrieval. It should also be noted that the current review is limited to the literature published in Web of Science, which may exclude some of the latest studies published in other types of documents that have not been published in Web of Science.

## 5. Conclusions

Through literature retrieval, bibliometric analysis, and keyword clustering, this paper screened 119 articles to describe the current application of network methods in CSOH, and then summarized its future directions, application bottlenecks, and research paradigm.

Future directions. From the perspective of theme words of clustering, eight clusters have hierarchical characteristics from macro to micro, which can be divided into four layers: key directions, core themes, key problems, and important methods. Cluster #1, “construction safety,” is the key research direction. Cluster #4, “safety management,” has gradually become the core theme of construction safety research based on network methods, showing the applicability of network methods in management research. Cluster #2, “human factors,” #5, “proactivity,” and #6, “construction sites,” reflect the core themes in the research on network methods in CSOH. Clusters #0, “Bayesian network,” #3, “data mining,” and #7, “artificial intelligence,” are important research methods closely related to the application of network methods in CSOH.

Application bottlenecks. The network approaches are not independently applied in CSOH research. It needs to rely on different theories (complex network theory, accident causation theory, communication theory, etc.) or be combined with other methods (NLP, artificial intelligence, IoT, etc.) and models (attribute-based building safety framework, epidemiological models SEM, BIM, etc.) to describe, model, and even reconstruct accident patterns and the contents of safety management systems. (1) The complex network theory and accident cause theory are the most used theoretical methods when using network methods for analyzing human error related to CSOH. (2) Considering the attributes and data characteristics of the research objects, the network model built based on an attribute-based building safety framework provides a more flexible definition and objective measurement of the association between research risks. When applied to CSOH, bottlenecks exist in the construction process of the network model in terms of data quality and calculation speed: (1) there are limitations to the quantity and quality of available data; (2) lack of practical verification opportunities has not been completely resolved.

Research paradigm. The research paradigm of network analysis process can be divided into four stages: description, explanation, prediction, and control. When the same network method encounters different research objects, it focuses on different analysis processes and plays different roles. Bayesian networks (BN and DBN) can be used in almost all engineering projects and to explore the relationship between unsafe behaviors and their influencing factors in human errors, labor, and personal issues. BN analysis focuses more on the description, explanation, and control measures of accident risk-influencing factors. DBN can conduct a predictive analysis by simulating the risk transmission mechanism. Neural networks (NN, ANN, FNN, and CNN) were applied to analyze the safety atmosphere, occupational health and safety, and safety behavior in construction safety management, especially in fall prevention. SNA can describe and explain the relationship between risks and unsafe behaviors and their influencing factors and has the potential of combining other models for dynamic prediction.

## Figures and Tables

**Figure 1 ijerph-19-12241-f001:**
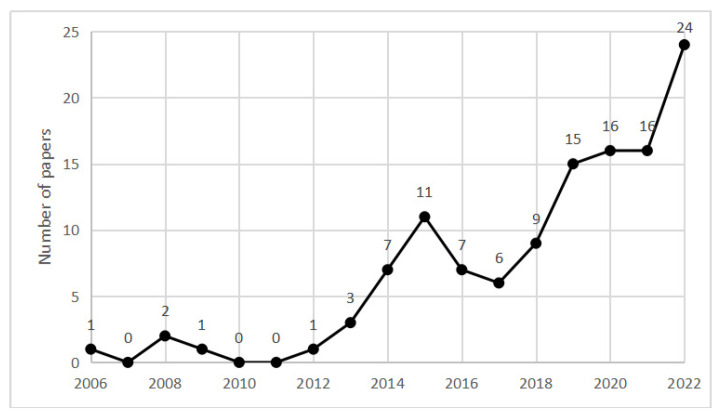
Numbers of papers from 2006 to 2022.

**Figure 2 ijerph-19-12241-f002:**
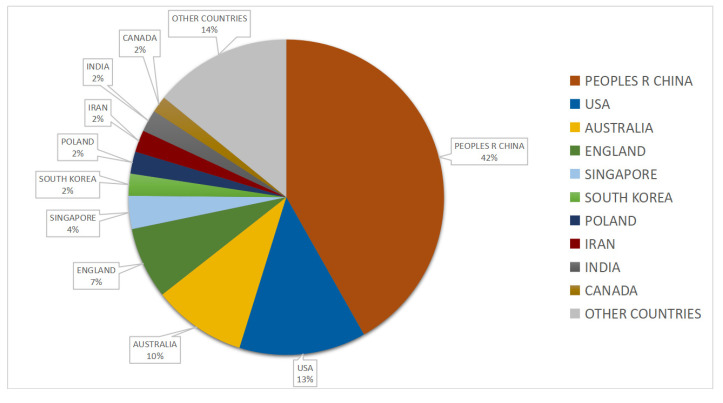
Distribution of countries (total number of articles: 119).

**Figure 3 ijerph-19-12241-f003:**
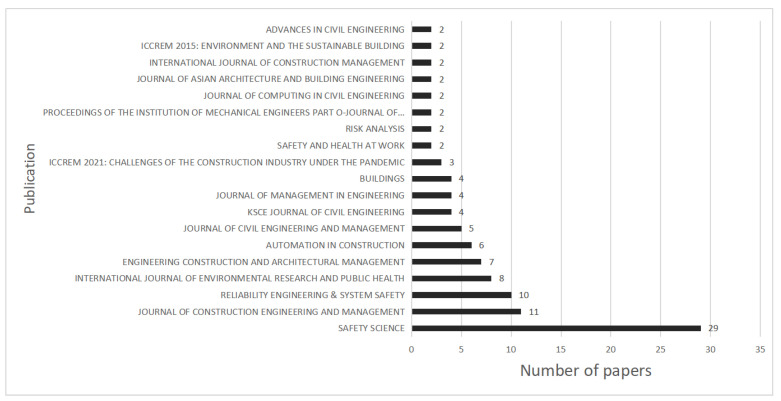
Numbers of papers published in major journals (n ≥ 2).

**Figure 4 ijerph-19-12241-f004:**
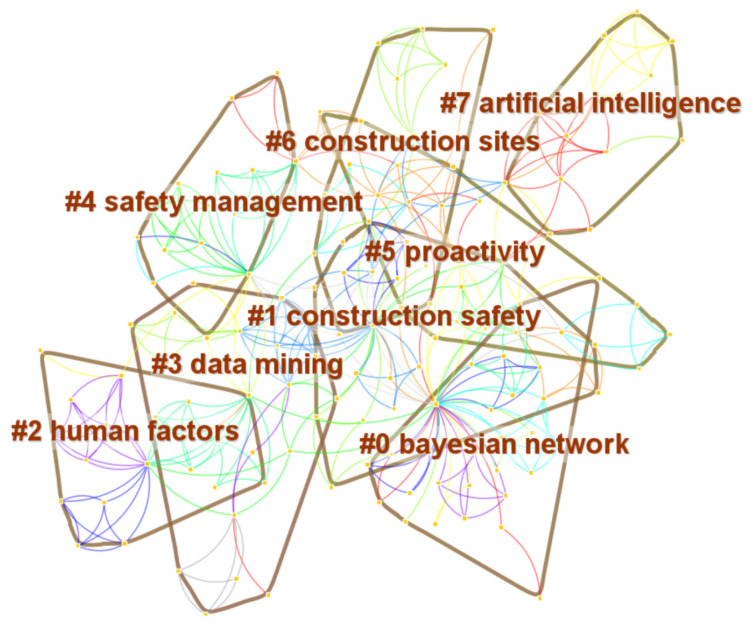
Network of keyword clusters.

**Table 1 ijerph-19-12241-t001:** List of search keywords used in Web of Science.

Number	Search Keywords
1	(health* OR accident OR injur*) AND (occupation* OR workplace OR worksite OR industry* OR construction* OR build* OR “civil engineering”)
2	(risk* OR danger* OR hazard* OR safety) AND management
3	Network

**Table 2 ijerph-19-12241-t002:** Distribution of institutions (total number of articles: 119).

Rank	Frequency (n ≥ 3)	Institution
1	11	Huazhong University of Science and Technology
2	7	Hong Kong Polytech University
3	6	Tsinghua University
4	5	University of Maryland
5	4	Polish Academy of Sciences
6	4	China University of Mining and Technology
7	4	China University of Geosciences
8	4	Birmingham City University
9	3	Georgia Institute of Technology
10	3	Curtin University
11	3	Memorial University of Newfoundland
12	3	Southeast University
13	3	North Carolina State University
14	3	Wuhan University of Technology
15	3	Nanyang Technological University
16	3	National University Singapore
17	3	Harbin Institute of Technology
18	3	Nanjing University of Aeronautics and Astronautics
19	3	Dalian University of Technology
20	3	City University of Hong Kong
21	3	Queensland University of Technology

**Table 3 ijerph-19-12241-t003:** Keyword Co-occurrence information.

Rank	Keywords	Frequency (n ≥ 3)	Co-Occurrence Keywords (Top 5)	Journals (Top 3)
1	bayesian network	21	construction safety (3); human error (3); accident prevention (2); safety management (2); fault tree (2)	Safety Science (8); International Journal of Environmental Research and Public Health (4); Reliability Engineering & System Safety (3)
2	construction safety	12	bayesian network (3); risk analysis (2); labor and personnel issues (2) safety climate (2); simulation (2)	Safety Science (3); Journal of Construction Engineering and Management (3); Reliability Engineering & System Safety (2)
3	safety management	8	network theory (3); accident analysis (2); electrical and mechanical (e&m) works (2); time series (2); construction (2)	Safety Science (4); Journal of Management in Engineering (1); Journal of Civil Engineering and Management (1)
4	accident analysis	6	safety management (2); electrical and mechanical (e&m) works (2); safety analysis (2); risk management (1); human factors (1)	Safety Science (3); International Journal of Environmental Research and Public Health (1); Reliability Engineering & System Safety (1)
5	complex network	6	unsafe behavior (2); accident prevention (1); safety management (1); risk interaction (1); construction workers (1)	Safety Science (2); International Journal of Environmental Research and Public Health (1); Reliability Engineering & System Safety (1)
6	construction management	5	accident prevention (1); bayesian network (1); artificial neural network(1); safety climate (1); risk management (1)	Safety Science (1); Reliability Engineering & System Safety (1); Journal of Management in Engineering (1)
7	construction industry	5	safety (2); bayesian network (1); social network analysis (1); human error (1); labor and personnel issues (1)	Safety Science (2); Journal of Construction Engineering and Management (1); Buildings (1)
8	accident prevention	5	behavioral risk chain (2); falls (1); complex network (1); risk assessment (1); probabilistic transmission path (1)	Safety Science (1); International Journal of Environmental Research and Public Health (1); Journal of Civil Engineering and Management (1)
9	labor and personnel issue	4	construction safety (2); neural network (2); safety climate (1); human error (1); data mining (1)	Journal of Construction Engineering and Management (4)
10	risk management	4	construction safety (1); construction management (1); accident analysis (1); simulation (1); bayesian network (1)	Engineering Construction and Architectural Management (2); Safety Science (1); Reliability Engineering & System Safety (1)
11	safety climate	4	construction safety (2); neural network (2); prediction (2); safety communication (1); safety management (1)	Safety Science (2); International Journal of Environmental Research and Public Health (1); Journal of Construction Engineering and Management (1)
12	human factor	4	accident analysis (1); safety assessment (1); hybrid approach (1); human reliability (1); safety analysis (1)	Safety Science (1); Reliability Engineering & System Safety (1); KSCE Journal of Civil Engineering (1)
13	artificial neural network	4	machine learning (2); artificial intelligence (2); safety management (1); safety climate (1); time series (1)	Automation in Construction (2); Journal of Management In Engineering (1); Safety and Health At Work (1)
14	data mining	3	neural network (2); construction safety (1); safety behavior (1); decision tree (1); neural network (1)	KSCE Journal of Civil Engineering (1); International Journal of Environmental Research and Public Health (1); Journal of Construction Engineering and Management (1)
15	resilience engineering	3	construction (1); safety management (1); human factors (1); safety management systems (1); super decisions software (1)	Safety Science (1); Reliability Engineering & System Safety (1); Safety and Health At Work (1)
16	neural network	3	safety climate (2); data mining (2); labor and personnel issues (2); prediction (2); construction safety (1)	Journal of Construction Engineering and Management (2); Safety Science (1)
17	risk assessment	3	bayesian network (2); accident prevention (1); construction management (1); bridge construction(1)	Reliability Engineering & System Safety(1); International Journal of Environmental Research and Public Health(1); Journal of Civil Engineering and Management (1)
18	fall risk	3	fault tree (2); bayesian network (2); bridge construction (1); steel construction (1)	Journal of Civil Engineering and Management (2); Safety Science (1)
19	construction site	3	construction safety (1); simulation (1); risk management (1); case study (1)	Journal of Civil Engineering and Management (1); Journal of Management In Engineering (1); Engineering Construction and Architectural Management (1)
20	unsafe behavior	3	complex network (2); accident prevention (1); behavioral risk chain (1); bayesian network (1); construction workers (1)	Journal of Construction Engineering and Management (1); Journal of Civil Engineering and Management (1); Engineering Construction and Architectural Management (1)
21	artificial intelligence	3	machine learning (2); artificial neural network (2); occupational health and safety (1); time series (1); safety management (1)	Automation in Construction (2); Proceedings of the Institution of Mechanical Engineers Part O—Journal of Risk and Reliability (1)

**Table 5 ijerph-19-12241-t005:** Network approaches, research objects, and analysis process.

Network Approaches	Research Objects	Analysis Process
ANN Artificial neural network	Occupational health and safety; construction [52]	explanation, prediction
	Safety climate; construction sites [54]	
	Near-miss falls; construction safety [26]	
	Construction occupational health and safety [55]	
BN Bayesian network	Safety culture; organizational culture; [56]	description, explanation, control
	Chains of unsafe behaviors; building construction; accident prevention [57]	
	Construction safety; safety management; human behavior; safety climate [58]	
	Prefabricated buildings; improved human factor analysis and classification system [59]	
	Operational tunnels [60]	
	Electrical and mechanical (E&M) works; accident analysis [61]	explanation, control
	Bridge construction; fall risks [19]	
	steel construction; fall risks [62]	
	Falling accidents; human-organizational factors [63]	description, explanation
	Construction safety; human error; labor and personnel issues [38]	
	Hydraulic engineering; human error [64]	
	Safety risk analysis; tunnel construction [65]	
	Occupational safety; accident prevention; Falls [18]	explanation, prediction
	Human error; construction industry [66]	
	Productivity; building project; construction management [67]	
DBN Dynamic Bayesian network	Occupational accidents; organization factors [68]	explanation, prediction
	Accident diagnosis and management [23]	
	Fall from height; construction workers [24]	
	Construction safety; predictive analysis; tunnel construction [28]	
CN Complex network	Human error; safety assessment [32]	explanation, prediction
	Near-miss; metro construction; safety management [42]	
	Construction safety; subway construction [25]	description, explanation
	Unsafe behaviors; accident prevention; urban railway construction [46]	
	Safety management; design for safety (DFS); prevention through design (PTD); subway construction [69]	
	Accident analysis; railway operational accident [70]	
	Accident analysis; metro operation hazard network (MOHN) [71]	
	Deep foundation pit; subway construction [17]	
	Construction workers; unsafe behavior [72]	
	Unsafe behavior; accident prevention; urban railway [73]	
	Accident level; accident chain; construction [44]	description, explanation, control
	Human factor analysis (HFA); occupational safety [48]	
	Organizational synchronization; construction delay factors [74]	
CNN Convolutional neural network	Fall prevention; personnel protective equipment [75]	explanation, prediction, control
	Construction safety; guardrail detection [29]	
FNN Fuzzy neural network	Worker-machine safety; intelligent assessment [76]	explanation, prediction, control
NN Neural network;	Cognitive analysis; safety behavior; working at height; labor and personnel issues [39]	explanation, control
	Construction hazard; site management; Ensemble Predictive Safety Risk Assessment [77]	explanation, prediction, control
Semantic network analysis	Urban infrastructure maintenance; user satisfaction [78]	description, explanation
SNA social network analysis	Safety management; construction projects; effective persons [79]	description, explanation, control
	Construction safety and health; ethnic minority workers [80]	
	Construction safety and health knowledge [81]	
	Green retrofit; stakeholders [82]	
	Construction industry; tower crane; collaborative governance [83]	
	Underground engineering; risk diffusion effect [84]	
	Communication; construction industry; labor and personnel issues [85]	
	Building information modeling; performance evaluation; construction management [86]	explanation, prediction, control
	Safety inspection; real-time association rules; proactive safety [36]	

## Data Availability

Data will be made available upon request from the corresponding author.

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
