# Peer review of "Research Paradigm of Network Approaches in Construction Safety and Occupational Health"

_ijerph, 2022, doi:10.3390/ijerph191912241_

Round 1

Reviewer 1 Report

In this paper, related studies in construction safety and occupational health (CSOH) are reviewed and the research paradigm of the recent decades is summarized. The authors solicited 125 peer-reviewed journal articles and performed bibliometric analysis and explored the research status, underpinning theories, major question categories, application paradigms, and development trends. Overall, the contents of the manuscript are well-organized and interesting. In my opinion, this paper could be recommended for publication in this Journal, but some sections of this manuscript are not clearly described and discussed. The reviewer suggests the manuscript be returned to the authors for major revisions:

1. Adding a time distribution map by year of publication in Section 3.1.1 and adding a distribution map by countries & institutions in Section 3.1.2 would be better.

2. Please improve the resolution of Figure 1.

3. Please improve the layout of Figure3, and use a better style for reader to read them clearly.  

4. The contents in Section 4 and Section 5 should be based on the results of the bibliometric analysis in Section 3. It is recommended to include the analysis on how derive the contents of Sections 4 and 5 through bibliometric analysis results.

5. The format of the references should be consistent. For example, the title case format of Reference 1 and Reference 2 is inconsistent. In addition, please check if Reference 2 and Reference 4 are duplicated.

Reviewer 2 Report

The paper provides a detailed review of research on network approaches to delineate their application areas, identify research gaps, and guide potential research directions in CSOH.

Exploring construction safety and occupational health (CSOH) is of great importance to transforming the high-risk, high-fatality environment of civil construction into a safe and healthy one. The authors presented critical issues worthy of discussion and comprehensive research examination. However, to emphasize the article's contribution and to understand its usefulness, I would like to highlight a few issues that need revision or better explanation.

1.      Introduction

i)      The first paragraph is a copy of an article template, which is not part of the content of this paper. Page 1, lines 28-36.

ii) Since the authors proposed the network approach, they should provide a brief clarification of this method.

2.      Literature retrieval

i)    In this chapter, a  flow of information through the different phases of a systematic Literature review is missing. It is advisable to map out the number of records identified, included, and excluded

3. Bibliometric analysis

i) The subchapters 3.1. and 3.1.2 lack some figures to refer to the data presented. t is advisable to show figures and graphs that support the numerical and textual data presented.

5. Network analysis from the view of description, explanation, prediction, and control

i)  It would be highly recommended that the authors in this chapter include a table summarizing the four categories analyzed (description, explanation, prediction, and control), the authors identified in the literature in each category, and the typologies of network approaches employed.

The article is well structured, and chapters 6 and 7 clearly emphasize its aims.

Reviewer 3 Report

Detailed comments in PDF markup.

Overall:

1. Be clear when you are discussing safety in general vs. for construction.  It gets confusing at times for a construction professional.

2. The "research" component of the paper is very generic.  Include more specific results with numbers, both in text and in graphics.  They exist, but the discussion was very very light in terms of hard evidence.  It felt a lot like "trust us this is what we saw."

3. With the addition of hard numbers and some revisions, this research can be extremely impactful for guiding future research efforts, and for getting industry to adopt them.

Reviewer 4 Report

1- The manuscript do not contains new and significant information to justify the publication and do not shows a relatively good progress in knowledge. In my opinion, the innovation of this manuscript is ambiguous.

2- In the introduction part, the existing work are described and referenced without a proper logic. A more targeted overview and summary according to the topic of this paper need to be conducted again.

The literature review seems is not be thoroughly done, so the paper is not very successful in elaborating the knowledge gap that the study is trying to fill. The flow of information on why this study is needed and how it builds on previous work and existing practice is unclear. Such a statement is very important for readers understanding, and should be presented clearly in Abstract, Introduction, and Conclusion. Specific to the literature review, the gap should be seen as the point of departure for the study. A revision is suggested to make the knowledge gap and the main contribution of the paper clear. Please discuss the limits of the literature and explain how you fill some of the existing gaps.

3- Figure 1 is not clear.

4- The methodology needs revisions and alone is not enough to analyze and develop the results and is not an innovation. Developing a hybrid model can provide better innovation. Reviewing and evaluating the selection and application of decision-making methods can be effective in providing a comprehensive hybrid model and provide better innovation. Also, the validation of the results requires more detailed descriptions and citations.

5- Conclusion section shall be improved indicating the findings in this article. In the current form, conclusion is very generic explanation. Try to be focused on what has been promised to deliver and what has been delivered. Conclusions should better reflect the paper contents and the results obtained in this article.

6- The results of the paper have a relatively practical process, but it is necessary for the authors to provide more explanations about the use of the results of different sections so that the audience can use it easily.

7- The language in the manuscript is weak and requires significant corrections to make it suitable for publication.

Reviewer 5 Report

Dear Authors,   the Article is scientific sounding, relevant and well-built.    However, in my opinion, there are some notes to be revised:
  1. The Results Section should with methodological results.
  2. The Discussion Section should estimate approaches, described in the Introduction Section. 
  3. The References Section should include more 2022 fresh items. 
Good luck with revisions!

Round 2

Reviewer 1 Report

The authors have addressed my comments.

Author Response

Thank you for your suggestions and recognition of this work.

Reviewer 5 Report

Dear Authors,

the article looks OK, but some English spelling check is needed.

Good luck!

Author Response

Thank you for your suggestions and recognition of this article. We checked the spelling errors carefully, for example:

In Line 350, we revised the "particully" to "particularly"

In Line 354, we revised the "safefy" to "safety"

In Line 423, we revised the "the most commenly used to" to " the most used"